# Structural and functional characterization of the mitochondrial complex IV assembly factor Coa6

Shadi Maghool[1] , N Dinesha G Cooray[1], David A Stroud[2] , David Aragão[3] , Michael T Ryan[4] , Megan J Maher[1,5]

**Assembly factors play key roles in the biogenesis of many multi-subunit protein complexes regulating their stability, activity, and the incorporation of essential cofactors. The human assembly factor Coa6 participates in the biogenesis of the $Cu_A$ site in complex IV (cytochrome $c$ oxidase, COX). Patients with mutations in Coa6 suffer from mitochondrial disease due to complex IV deficiency. Here, we present the crystal structures of human Coa6 and the pathogenic [W59C]Coa6-mutant protein. These structures show that Coa6 has a 3-helical bundle structure, with the first 2 helices tethered by disulfide bonds, one of which likely provides the copper-binding site. Disulfide-mediated oligomerization of the [W59C]Coa6 protein provides a structural explanation for the loss-of-function mutation.**

## Introduction

The mitochondrial oxidative phosphorylation (OXPHOS) system generates the bulk of cellular ATP, fuelling the energy demands of most eukaryotes. Five multi-subunit protein complexes in the mitochondrial inner membrane, termed complexes I–V, comprise the OXPHOS system. Complex IV (cytochrome $c$ oxidase; COX) is the last complex of the electron transport chain, transferring electrons from cytochrome $c$ to molecular oxygen, and in the process, pumping four protons across the inner membrane [1]. In mammals, complex IV is additionally found with complexes I and III in large multicomplex assemblies termed the respiratory chain super-complexes [2, 3]. In humans, complex IV is composed of 14 subunits with the three mitochondrial DNA-encoded subunits (COX1-3) forming the catalytic core of the enzyme that is conserved from yeast to man [4]. Cytochrome $c$ docks onto the intermembrane space (IMS) domain of COX2, which contains a binuclear copper center, termed $Cu_A$ that accepts the electrons. The electrons are then passed to a heme $a$ group in COX1 and then to a heme $a_3$-$Cu_B$

site and finally to oxygen, which is reduced to water [5]. Thus, reduction of the $Cu_A$ site in COX2 represents the critical first stage in complex IV activity.

In humans, the assembly of the $Cu_A$ site requires the assembly factors Cox16 [6], Cox17 [7], Sco1 [8], Sco2 [9], and Coa6 [10, 11], with mutations in these proteins resulting in COX assembly defects and an attenuation in the activity of the complex [12, 13]. The metallochaperone Cox17 [7] is a mitochondrial IMS located protein, which receives copper from the mitochondrial matrix copper pool where copper is bound by an anionic fluorescent molecule (also known as copper ligand, CuL) [8, 14, 15]. CuL has also been reported to be located in the cytoplasm and may be a vehicle for copper transport into mitochondria [14]. Cox17 is crucial for copper delivery to COX2 through a proposed sequential pathway, with Sco2 acting upstream of Sco1 [16, 17, 18, 19]. Recently, Cox16 has been shown to play a role in this process, being required for the association between Sco1 and COX2 and between COX1 and COX2 for COX assembly [6].

Coa6 is a soluble IMS protein with a $CX_9C$–$CX_{10}C$ sequence motif, which binds Cu(I) with a $K_{D(CuI)}$ of ~$10^{-17}$ M [11]. Coa6 deletion in both yeast and human cells results in diminished COX assembly and activity; however, this defect can be partially rescued by exogenous copper supplementation [10] and in yeast by treatment with elesclomol (via a proposed elesclomol–copper complex [20]). Recent studies [11, 21] have shown that Coa6 interacts with newly synthesized COX2 [11, 21] and with both Sco1 and Sco2, leading to suggestions that Coa6 acts to facilitate $Cu_A$ site assembly and, therefore, COX biogenesis as part of a Sco1/Sco2 [21]-containing complex and/or through the redox cycling of Sco2 [22].

Pathogenic mutations in the Coa6 protein (W59C and E87*) were identified in a patient suffering from hypertrophic obstructive cardiomyopathy, resulting in a COX defect in the heart tissue, but no defect in fibroblasts [17]. An additional patient with a W66R mutation in Coa6 suffered from neonatal hypertrophic cardiomyopathy, muscular hypotonia, and lactic acidosis with a COX defect in the fibroblasts [16]. One report examining the [W59C]Coa6 variant showed the protein was mistargeted to the mitochondrial matrix; however, our previous results [11] showed that the [W59C]Coa6 was

[1]Department of Biochemistry and Genetics, La Trobe Institute for Molecular Science, La Trobe University, Melbourne, Australia [2]Department of Biochemistry and Molecular Biology and The Bio21 Molecular Science and Biotechnology Institute, The University of Melbourne, Parkville, Australia [3]Australian Synchrotron, Australian Nuclear Science and Technology Organisation, Clayton, Australia [4]Department of Biochemistry and Molecular Biology, Biomedicine Discovery Institute, Monash University, Clayton, Australia [5]School of Chemistry and The Bio21 Molecular Science and Biotechnology Institute, The University of Melbourne, Parkville, Australia

Correspondence: megan.maher@unimelb.edu.au
David Aragão's present address is Diamond Light Source, Harwell Science and Innovation Campus, Didcot, UK

able to partially rescue human embryonic kidney (HEK293T) Coa6 knockout cells, suggesting that the mutant protein retains some functionality.

It is clear that Coa6 is essential for COX assembly, that it plays a role in the biogenesis of the $Cu_A$ site, interacts with crucial factors in the IMS Cu delivery pathway, and that mutations to Coa6 are pathogenic and lead to mitochondrial disease ([16], [17]). However, the precise role that Coa6 plays in COX biogenesis, and therefore, the mechanism underlying [W59C]Coa6 pathogenesis remains to be elucidated. In this study, we have determined the crystal structures of the human wild-type Coa6 ([WT]Coa6) and [W59C]Coa6 proteins by X-ray crystallography to 1.65 and 2.2 Å resolution, respectively. By mutagenesis, we propose the location of the copper-binding site within Coa6. Examination of the structures allows us to suggest the mechanism of action for this protein in COX assembly and the molecular origin of pathogenesis for the W59C mutation.

# Results and Discussion

## The structure of [WT]Coa6

Recombinant [WT]Coa6 protein was overexpressed in *Escherichia coli* strain SHuffle T7, which promotes the production of correctly disulfide bonded active proteins within the cytoplasm ([23], [24]). The fully oxidized [WT]Coa6 protein (including two disulfide bonds) was purified by affinity and size-exclusion chromatography (SEC). The redox state of the purified [WT]Coa6 protein was confirmed by Ellman's assay ([25]). To elucidate the atomic structure of the [WT]Coa6 protein, we crystallized and determined its structure to 1.65 Å resolution by X-ray crystallography (Table 1). The structure shows two monomers of [WT]Coa6 per asymmetric unit, arranged as an antiparallel dimer (Fig 1A). The final [WT]Coa6 model includes residues 52–111 from molecule A (Fig 1A, cyan) and residues 50–119 from

**Table 1. Data collection and refinement statistics.[a]**

| Data collection | | | |
|---|---|---|---|
| Crystal | Native [WT]Coa6 | Anomalous [WT]Coa6 | [W59C]Coa6 |
| Wavelength (Å) | 0.9918 | 1.722 | 0.954 |
| Temperature (K) | 100 | | |
| Diffraction source | Australian synchrotron (MX2) | | |
| Detector | ADSC quantum 315r | | |
| Space group | $P2_12_12_1$ | $P2_12_12_1$ | $P1$ |
| $a, b, c$ (Å) | 32.0, 52.4, 78.3 | 32.2, 52.7, 78.6 | 41.4, 47.8, 48.1 |
| $\alpha, \beta, \gamma$ (°) | 90, 90, 90 | 90, 90, 90 | 116.9, 98.8, 104.1 |
| Resolution range (Å) | 50.0–1.65 (1.71–1.65) | 44.0–2.28 (2.35–2.28) | 50.0–2.20 (2.28–2.20) |
| Total no. of reflections | 80,326 | 416,239 | 49,280 |
| No. of unique reflections | 16,393 | 6,519 | 15,400 |
| Completeness (%) | 99.8 (100.0) | 99.5 (95.1) | 98.4 (97.3) |
| Redundancy | 4.9 (5.0) | 63.9 (33.7) | 3.2 (3.1) |
| $\langle I/\sigma(I) \rangle$ | 29.9 (2.3) | 70.3 (5.6) | 14.4 (2.1) |
| $R$merge (%) | 4.7 (67.3) | 5.8 (73.1) | 5.9 (59.6) |
| $R$pim (%) | 2.3 (32.8) | 0.7 (12.4) | 4.1 (40.8) |
| Refinement statistics | | | |
| Resolution range (Å) | 43.6–1.65 (1.69–1.65) | | 38.3–2.2 (2.129–2.185) |
| No. of reflections, working set | 14,716 | | 14,685 |
| No. of reflections, test set | 1,632 | | 713 |
| $R_{work}$ (%) | 18.8 (27.5) | | 20.0 (27.3) |
| $R_{free}$ (%) | 22.9 (33.4) | | 24.4 (37.2) |
| R.m.s.d. bond lengths (Å) | 0.012 | | 0.003 |
| R.m.s.d. bond angles (°) | 1.75 | | 1.21 |
| Ramachandran[b] | | | |
| Favored, % | 100 | | 98.3 |
| Allowed, % | 100 | | 100 |
| PDB ID code | 6PCE | | 6PCF |

[a]Values in parenthesis are for highest resolution shell.
[b]Calculated using MolProbity ([67]).

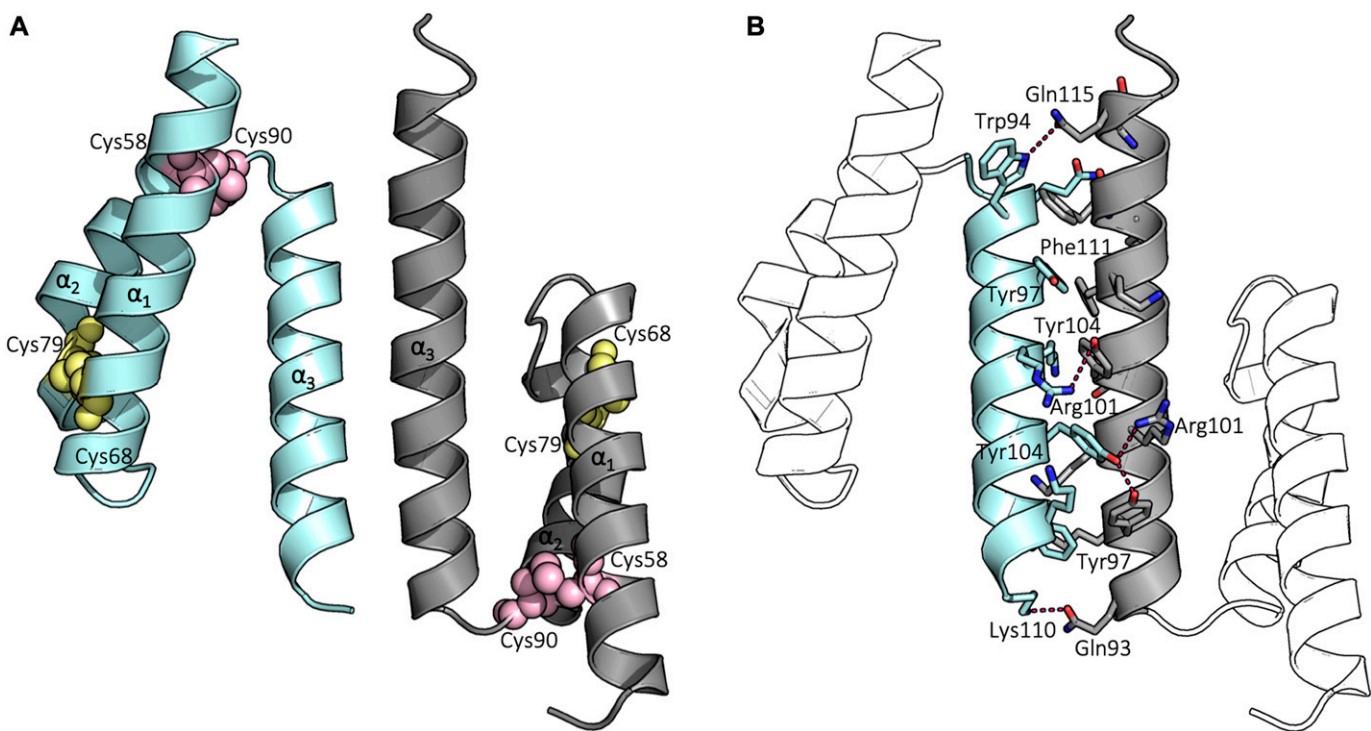

**Figure 1. Structure of the $^{WT}$Coa6.**
**(A)** Cartoon representation of the overall structure of $^{WT}$Coa6. Secondary structures are represented as cartoons with monomers colored in cyan and gray. Residues Cys58 and Cys90 are shown as pink spheres, whereas Cys68 and Cys79 residues are shown as yellow spheres. **(B)** Residues located at the dimer interface are shown as sticks and hydrogen bonds between residues (labeled) shown as dashed lines.

molecule B (Fig 1A, gray). Interpretable electron density was not observed for residues A47–51, A112–125, B47–49, and B120–125, and these were omitted from the final model. The superposition of molecule A with molecule B gives a root mean square deviation (r.m.s.d.) for 53 common Cα positions of 0.43 Å, indicating the structures of the two monomers are very similar.

Each monomer is composed of a 3-helical bundle, with a coiled–coil–helix–coiled–coil–helix (CHCH) fold (26) (Fig 2C). The sequence of $^{WT}$Coa6 includes four Cys residues (at positions 58, 68, 79, and 90) in a $CX_9C–CX_{10}C$ sequence motif, which in the crystal structure form two intramolecular disulfide bonds per monomer, between Cys58–Cys90 and Cys68–Cys79, in accordance with the fully oxidized state of the protein (Fig 1A). These disulfides tether helices $\alpha_1$ and $\alpha_2$ together at each end of the helical pair.

Proteins which include twin $CX_9C$ motifs and adopt CHCH folds include the mitochondrial copper chaperone Cox17 (7) (Fig 2A, Protein Data Bank [PDB] 2RNB), Mia40 (PDB 2K3J) (27), CMC4 (p8MTCP1; PDB 1HP8) (28), CHCHD5 (PDB 2LQL), and CHCHD7 (PDB 2LQT) (29), whose structures have been elucidated by Nuclear Magnetic Resonance. In addition, a search of the $^{WT}$Coa6 co-ordinates against the PDB revealed (30) that $^{WT}$Coa6 shares significant structural similarity with the Cox6B subunit of COX (Fig 2B, PDB 2ZXW (31), Chains H and U, r.m.s.d. 1.8 Å for 51 common Cα positions). The structure of Cox6B also shows a CHCH fold and includes four Cys residues, which are found in a $CX_9C–CX_{10}C$ sequence motif. These Cys residues also form two pairs of intramolecular disulfide bonds (between Cys29–Cys64 and Cys39–Cys53),

with positions that superpose exactly with the $^{WT}$Coa6 structure (Figs 2B and S1). Because of its CHCH structure, Cox6B has been included in the twin Cx9C protein family (despite showing a sequence variation: $CX_9C–CX_{10}C$). Coa6 can now also be included in that family. All structurally characterized twin $CX_9C$ proteins listed here include an N-terminal ($\alpha_1/\alpha_2$) helical pair, which is tethered at each end by disulfide bonds (Fig 2) and most have been shown to undergo import into the mitochondrial IMS through an Mia40-mediated oxidative folding pathway (26, 32, 33). However, only the Cox17 protein has been shown to bind Cu(I) (7) and to play a role in COX assembly.

The Coa6 dimer has buried surface areas of 701 and 656 Å$^2$ (for molecules A and B, respectively), which are 15 and 13%, of the total surface areas of each corresponding monomer, indicating that $^{WT}$Coa6 forms a stable dimer in solution (34). This agrees with our previous SEC data, which demonstrated that $^{WT}$Coa6 eluted from an analytical column at a volume corresponding to the molecular weight of a dimer (11). The Coa6 dimer is entirely mediated by intermolecular contacts between Helix $\alpha_3$ from each monomer. These contacts include electrostatic interactions, including hydrogen bonds and salt bridges (Fig 1B). In addition, Tyr97 of monomer A (Helix $\alpha_3$, cyan, Fig 1B) forms a $\pi$-stacking interaction with Phe111 of monomer B (Helix $\alpha_3$, gray, Fig 1B) at the dimer interface.

To investigate the roles of individual interface residues in the stabilization of the dimer, we generated variant proteins where interface residues were mutated to alanine (Y97A, Y104A, R101A, and Y97A/Y104A mutant proteins) and analyzed the quaternary structures

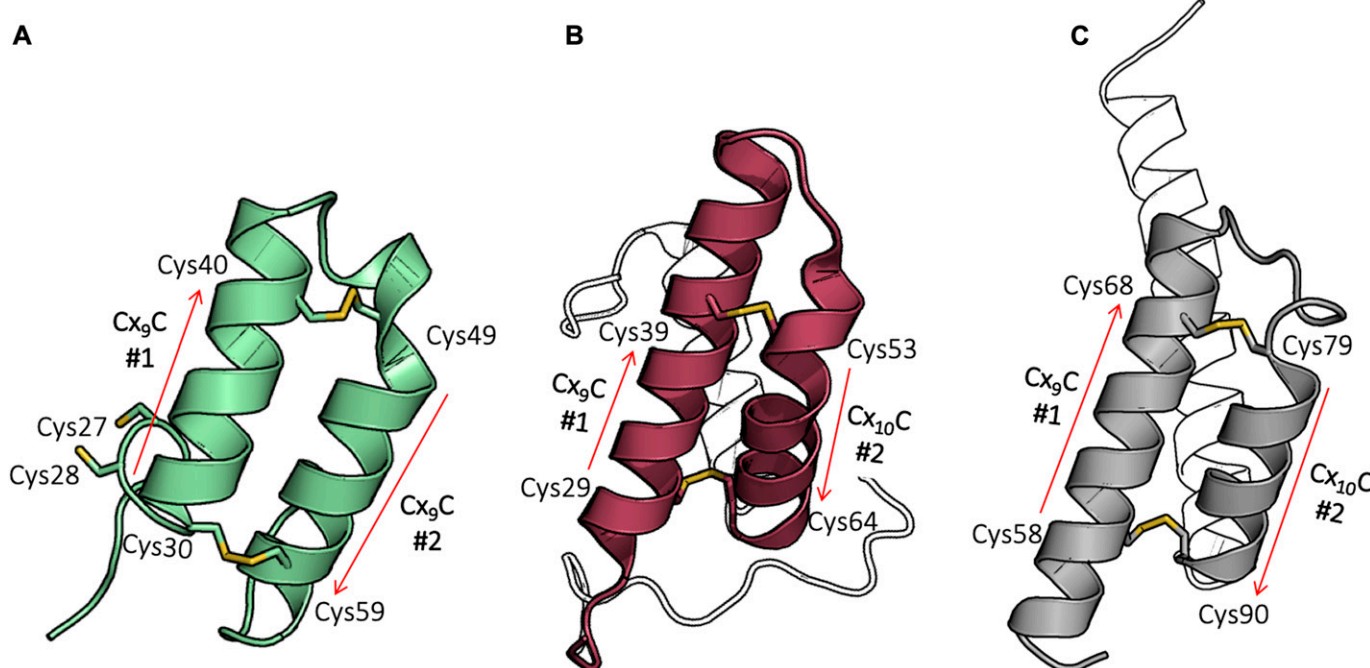

**Figure 2. $^{WT}$Coa6 has a twin CX$_9$C protein fold.**
**(A)** Cox17 (PDB 2RNB): helices are shown in green, and cysteines in the CX$_9$C motifs that form disulfide bonds are shown as yellow sticks. **(B)** Cox6B (PDB 2EIJ): helices are shown in raspberry, and cysteines in the CX$_9$C (or CX$_{10}$C) motifs that form disulfide bonds are shown as yellow sticks. **(C)** $^{WT}$Coa6 (this work): helices are shown in gray, and cysteines in the CX$_9$C (or CX$_{10}$C) motifs that form disulfide bonds are shown as yellow sticks. In all panels, the positions of the corresponding motifs are shown as red arrows. In panel (B), part of the N terminus and helix $\alpha_3$ in panels (B) and (C) are not colored for clarity.

of these variant proteins by analytical SEC. All mutant proteins eluted from SEC at volumes similar to that for dimeric $^{WT}$Coa6 (Fig S2), indicating conservation of their dimeric quaternary structures and therefore, the extensive nature of the dimer interface. Interestingly, no other twin CX$_9$C proteins have been characterized as dimeric and all reported structures to date have shown monomeric proteins (7, 27, 28, 29).

### Mutation of residues Cys58 and Cys90 eliminates Cu(I) binding to Coa6

We have previously reported that both the $^{WT}$Coa6 and $^{W59C}$Coa6 proteins bind Cu(I) with $K_{DCu(I)} \sim 10^{-17}$ M (11). To determine the stoichiometry of Cu(I) binding, purified apo-$^{WT}$Coa6 was loaded with Cu(I) in the presence of the reducing agent as previously described (11, 35). Following SEC to remove excess Cu, $^{WT}$Coa6 was found to bind Cu(I) with a Cu:protein stoichiometry of 1:1 (Fig S3), indicating the presence of a single Cu(I) binding site per molecule of $^{WT}$Coa6.

CX$_n$C motifs are commonly present in soluble, high-affinity Cu(I)-binding proteins (copper metallochaperones) such as Atox1 (36), where redox cycling of the pair of Cys residues from oxidized (disulfide, S–S) to reduced (2SH) allows Cu(I) co-ordination between the thiolate groups of the Cys side chains. CHCH fold proteins have been characterized with redox active disulfide bonds; however, this activity is mediated not by the Cys residues that are part of the CX$_9$C motifs but by additional pairs of Cys residues at the N termini of the proteins (37). For example, in

Cox17, an N-terminal C–C motif shows oxidoreductase activity and also binds Cu(I) with femtomolar affinity, for transfer to receiving copper proteins of the IMS, such as Cox11, Sco1, and Sco2 (8, 38, 39). Mia40 has a conserved N-terminal CPC motif that catalyzes the formation of intermolecular disulfides in CX$_9$C proteins within the IMS (40). The redox potentials of the N-terminal Cys pairs for Cox17 and Mia40 are –198 (41) and –200 mV (27). In addition, the human Sco1 and Sco2 proteins have been shown to possess redox-active disulfide bonds that bind Cu(I) when in the reduced state, with measured redox potentials for the respective S–S/2SH couples of –277 and less than –300 mV (9, 22, 38).

Our previous study showed that $^{WT}$Coa6 exists in mitochondria in a partially reduced state, that is, with one of the two intramolecular disulfide bonds reduced (11). Reduction of one disulfide bond would facilitate Cu(I) binding and yield Cu:protein stoichiometry of 1:1. We therefore sought to determine whether a Cu(I) binding site might exist between the Cys58–Cys90 or Cys68–Cys79 disulfides in $^{WT}$Coa6. To that end, we determined the redox potential of the S–S/ 2SH couple for the Coa6 protein (using a DTT$_{Red}$/DTT$_{Ox}$ redox couple) as –349 ± 1 mV (pH 7.0, Fig S4). To determine whether this represented one or both of the S–S/2SH redox couples in $^{WT}$Coa6 (i.e., the Cys58–Cys90 and/or the Cys68–Cys79 disulfides), the purified protein was incubated with DTT$_{Red}$/DTT$_{Ox}$ (40:1), followed by labeling with iodoacetamide (IAA) before analyses by both Matrix Assisted Laser Desorption/Ionization-Time of Flight (MALDI-TOF) mass spectrometry (MS) (Fig S5 and Table S1) and tryptic digest and peptide analysis by tandem MS/MS (Table 2). The MALDI-TOF data revealed that the dominant species in the reduced $^{WT}$Coa6 sample

**Table 2.** $^{WT}$Coa6 mass determined via MS/MS following reduction and IAA labeling.

| Peptide sequence | Predicted mass of peptide sequence (D) | Peptide position | Residue | Predicted mass upon alkylation (D) | Determined mass upon alkylation (D) |
|---|---|---|---|---|---|
| FEAGQFEPSETTAK | 1541.7118 | 111–124 | — | — | ND |
| SSFESS**C$_{90}$**PQQWIK | 1526.6944 | 84–96 | Cys90 | 1584.6999 | 1584.8 |
| C$_{68}$LDENLEDASQC$_{79}$K | 1467.6090 | 68–80 | Cys68, Cys79 | 1583.6199 | ND |
| GPLGSMAAPSMK | 1146.5645 | 42–53 | — | — | ND |
| QV**C$_{58}$**WGAR | 819.3930 | 56–62 | Cys58 | 877.3985 | 877.4 |
| DEYWK | 740.3250 | 63–67 | — | — | ND |
| YFDK | 572.2715 | 97–100 | — | — | ND |
| DYLK | 538.2871 | 103–106 | — | — | ND |

The determined masses of peptides, including Cys58 and Cys90, were detected at the predicted molecular masses of the peptide–IAA adducts, which were calculated by PEPTIDEMASS (57), indicating the reduction of the Cys58–Cys90 disulfide bond under these conditions. The alkylated residues Cys58 and Cys90 are highlighted in bold. ND, not detected.

included two free thiol groups (2SH [not 4SH], Fig S5 and Table S1). The MS/MS data showed that peptides that included residues Cys58 and Cys90 yielded molecular weights corresponding to adducts with IAA, indicating reduction of the Cys58–Cys90 disulfide bond under these conditions (Table 2).

In addition, we generated $^{C58S/C90S}$Coa6 and $^{C68S/C79S}$Coa6 variants, which were analyzed for Cu(I) binding as previously described (11). Cu(I)-binding to the $^{C58S/C90S}$Coa6-mutant protein was not detected using this assay (Fig 3). In contrast, Cu(I) binding to the $^{C68S/C79S}$Coa6 variant gave a measured $K_{D(Cu(I))}$ ~10$^{-16}$ M, which is ~10-fold weaker than that determined for the $^{WT}$Coa6 protein (11). Circular dichroism (CD) spectroscopic analyses of the mutant proteins showed that both proteins gave spectra similar to that of the $^{WT}$Coa6 protein, indicating that the observed changes in Cu(I)-binding properties were not due to alterations in the secondary structures of the proteins in the presence of the

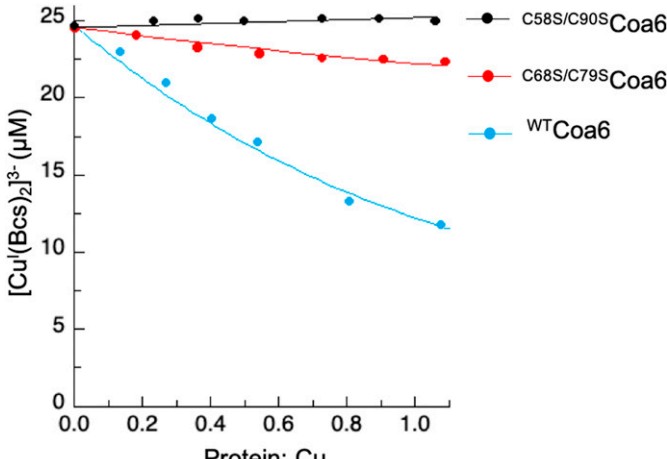

**Figure 3.** Determination of $K_D$ for Cu(I) binding for the $^{WT}$Coa6, $^{C58S/C90S}$Coa6, and $^{C68S/C79S}$Coa6 proteins.
The data were analyzed via a plot of the concentration of the [Cu$^I$(BCS)$_2$]$^{3-}$ complex (calculated from the absorption values at 483 nm versus the protein:Cu ratio) and the data fit using the equation previously described (11, 35).

mutations (Fig S6). However, the possibility exists that the mutagenesis affected the flexibility of the protein and/or its tertiary structure as observed for a similar analysis of the human Cox17 protein (42).

Interestingly, an analysis of the geometries and potential energies of the Cys58–Cys90 and Cys68–Cys79 disulfides (43) revealed that the Cys58–Cys90 bond has a +/−RHHook geometry, which is commonly observed for disulfide bonds within thioredoxin-like proteins,beeing classified as a "catalytic" conformation. These types of disulfide bonds are observed to redox cycle, which is the foundation of their activities. This is in comparison with the geometry of the Cys68–Cys79 disulfide which shows a −LHSpiral geometry, which is the geometry commonly observed for structurally stable disulfide bonds (43, 44, 45). In addition, the structure of $^{WT}$Coa6 shows that the Cys58–Cys90 site is proximate to positively charged residues (Lys53 and Arg55) and bordered by aromatic side chains (Trp59 and Trp94), which partially shield the site from the solvent (Fig S7). Positively charged residues in the vicinity of Cu(I)-binding sites within structurally characterized metallochaperone proteins such as Atox1 have been proposed to stabilize Cu(I) binding and mediate its transfer to other protein partners (46, 47). In addition, cation-$\pi$ interactions (between Cu(I) and aromatic amino acid side chains), such as those observed in the crystal structure of the Cu(I)-CusF protein from *E. coli* (48), are known to stabilize copper binding.

To further investigate the position of the Cu(I)-binding site, we expressed the FLAG-tagged $^{WT}$Coa6 and FLAG-tagged $^{C58S/C90S}$Coa6 double mutant in COA6$^{KO}$ cells and compared the levels of assembled COX with control HEK293T and COA6$^{KO}$ cells by blue native-PAGE (BN-PAGE) and Western blotting. As previously reported (11), the complementation with $^{WT}$Coa6-FLAG in COA6$^{KO}$ cells restored COX assembly. However, complementation with $^{C58S/C90S}$Coa6-FLAG was not able to rescue COX assembly and activity in COA6$^{KO}$ cells. Examination of the cellular localization of the overexpressed $^{C58S/C90S}$Coa6-FLAG protein by immunofluorescence showed that unlike $^{WT}$Coa6-FLAG, the $^{C58S/C90S}$Coa6 protein was cytosolic (Fig S8). CX$_9$C–CX$_{10}$C and CX$_9$C proteins require the presence of the Cys residues within these motifs for

import into the IMS, and oxidative folding via Mia40 to trap the proteins in the IMS (49, 50). The absence of IMS localization of the [C58S/C90S]Coa6-FLAG protein correlates with these observations and impeded our efforts to probe the role of these residues in Coa6 function and specifically COX assembly in human cells. Interestingly, however, an examination of the [C25A]yCoa6 and [C68A]yCoa6 (equivalent to human [C90A]Coa6) variants from yeast, showed that unlike [WT]yCoa6, these mutants were not able to rescue the respiratory growth defect of COA6[KO] cells (10). In addition, these mutations were shown to disrupt the interactions (probed by co-immunoprecipitation) between yCoa6 and yCox2 and ySco1 (51).

Taken together, the redox potential of the [WT]Coa6 Cys58–Cys90 disulfide, the results of mutagenesis, Cu(I)-binding experiments, and analyses of the geometries of the intramolecular disulfide bonds of [WT]Coa6 indicate that the [WT]Coa6 Cys58–Cys90 disulfide may redox cycle and in the reduced state (2SH), bind Cu(I). The redox potential of the [WT]Coa6 Cys58–Cys90 S–S/2SH redox couple at

−349 ± 1 mV indicates [WT]Coa6 could reduce the disulfide bonds in COX2 (−288 ± 3 mV) (22), and Sco1 (−277 ± 3 mV) (38) and could either reduce or be reduced by Sco2 (less than −300 mV) (9). In fact, a recent examination of this pathway suggested [WT]Coa6 may play such a role (22). Unfortunately, despite extensive attempts, we were unable to crystallize the Cu(I)-bound [WT]Coa6. Confirmation of Cu(I) binding and the atomic details of the Cu(I) site structure, therefore, await future investigation.

## The structure of [W59C]Coa6 reveals a disulfide-mediated oligomerization

Finally, we determined and refined the crystal structure of the pathogenic mutant protein [W59C]Coa6 to 2.2 Å resolution by X-ray crystallography (Table 1). The structure of [W59C]Coa6 shows four molecules (A, B, C, and D) per asymmetric unit. The noncovalent dimer observed in the [WT]Coa6 structure (Fig 4A) is maintained for [W59C]Coa6 (between molecules A [cyan] and B [gray]) through

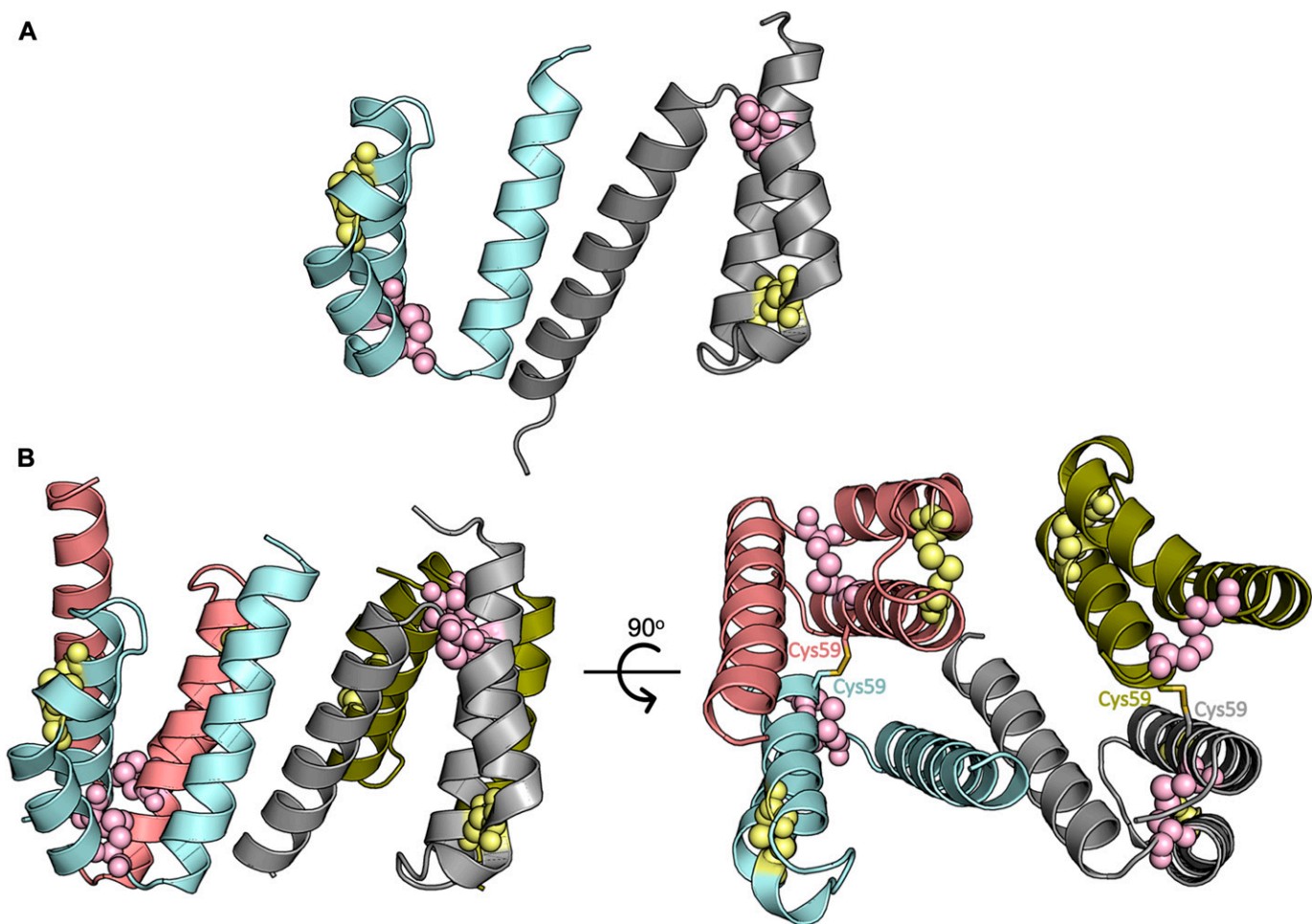

**Figure 4. Structure of the [W59C]Coa6-mutant protein.**
**(A)** Cartoon representation of the overall structure of [WT]Coa6. Secondary structures are represented as cartoons with monomers colored in cyan and gray. **(B)** Cartoon representation of the overall structure of [W59C]Coa6. Secondary structures are represented as cartoons with monomers colored in cyan, gray, salmon and gold. Residues Cys58 and Cys90 are shown as pink spheres and residues Cys68 and Cys79 as yellow spheres. Each monomer (cyan and gray) is linked to another monomer (salmon and gold, respectively) by an intermolecular disulfide bond (shown as yellow sticks) through the introduced Cys59 residue.

contacts along Helix $\alpha_3$ from each monomer (Fig 4B). In addition, each of these molecules form a covalent dimer with another molecule of $^{W59C}$Coa6 through a disulfide bond between the introduced residue Cys59 on each chain (covalent dimers between molecules A/D, [cyan and salmon] and B/C [gray and gold], Fig 4B). In this way, the asymmetric unit is composed of a dimer of dimers (a noncovalent dimer of disulfide bridged dimers). Importantly, this pattern of interactions is propagated throughout the crystal so that every molecule of $^{W59C}$Coa6 interacts with two others—one through noncovalent contacts along Helix $\alpha_3$ (as observed for $^{WT}$Coa6, cyan and gray, Fig 4B) and the other through a disulfide bond between residue Cys59 on each molecule as a result of the mutation (cyan/salmon and gray/gold, Fig 4B).

The superposition of the $^{WT}$Coa6 structure with the noncovalent A/B dimer of $^{W59C}$Coa6 yields a r.m.s.d. value of 1.2 Å for 113 common C$\alpha$ atoms. These minor structural differences originate from a slightly different association between the monomers in the dimers, rather than specific structural rearrangements. Importantly, in the $^{WT}$Coa6 and $^{W59C}$Coa6 structures, the positions of the proposed Cys58–Cys90 Cu(I)-binding sites are identical. Because the Cu(I)-binding activity of the $^{W59C}$Coa6 protein is preserved in the presence of the mutation (11), these residues, being positioned at the "end" of the ($\alpha_1/\alpha_2$) helical pair presumably remain accessible to Cu(I) binding, despite the oligomerization of the $^{W59C}$Coa6.

Our previous analysis of the quaternary structure of the $^{W59C}$Coa6-mutant protein by SEC (11) showed a broad elution profile, indicating protein oligomerization. Under reducing conditions, the elution profile for the $^{W59C}$Coa6 protein was identical to that of $^{WT}$Coa6, corresponding to a dimeric protein (11). These observations agree with the crystal structure described here, which shows that $^{W59C}$Coa6 oligomerization is mediated by the presence of intermolecular disulfide bonds that bridge noncovalent $^{W59C}$Coa6 dimers.

Characterizations of the $^{W59C}$Coa6-mutant protein in yeast ($^{W26C}$yCoa6) and human cells by other groups have suggested differing localizations of the mature protein: to the IMS (51) and mislocalization to the matrix (in U2OS cells (21)). Our previous study showed that complementation of $^{W59C}$Coa6 in COA6$^{KO}$ HEK293T cells only partially restored COX assembly and that in yeast, accumulation of yCoa6 in the mitochondrial IMS was impaired in the presence of the mutation W26C (equivalent to human $^{W59C}$Coa6) (11). Additional studies have also shown a decrease in steady state levels of Coa6 in yeast for the $^{W26C}$yCoa6-mutant protein (10) and an impaired interaction of the mutant protein with COX2 (51).

The structure of $^{W59C}$Coa6 is consistent with the bulk of these data. Oligomerization of the protein, through the creation of intermolecular disulfide bonds may inhibit or eliminate targeted protein–protein interactions between Coa6 and proteins such as Sco1, Sco2, and COX2, which are critical for its function. Certainly, the structures and charge distributions on the surfaces of the $^{WT}$Coa6 dimer and $^{W59C}$Coa6 tetramer are different (Fig S9A and B). In particular, condensed areas of positive charge on the surface of the $^{WT}$Coa6 dimer (Fig S9A) are not present in the $^{W59C}$Coa6 tetramer (Fig S9B). These regions may mediate interactions with the

Sco1, Sco2, and COX2 proteins through concentrated areas of negative charge on the surface structures of those proteins (Fig S9C–E).

## Conclusion

Despite intense investigation, the identities of all proteins involved, their respective roles, and the sequence of their participation in complex IV biogenesis are currently not known (4, 22, 52, 53). The COX assembly factor Coa6 was identified through a proteomic survey of the mitochondrial IMS from *Saccharomyces cerevisiae* (49) and the significant number of investigations that followed, proposed its role in the biogenesis of the Cu$_A$ site, presumably through the interaction of Coa6 with other critical proteins in the COX biogenesis pathway such as Sco1 and Sco2 (10, 11, 21, 51). However, the molecular details of how Coa6 functions in this pathway, particularly the molecular foundations of its ability to bind copper have until now been undefined. Crucially, the molecular basis of the pathogenic effect of the $^{W59C}$Coa6 mutation has remained elusive.

Here, through the determination of the high-resolution crystal structures of the $^{WT}$Coa6 and $^{W59C}$Coa6 proteins, we have contributed to the interpretation of the role of Coa6 in COX biogenesis. The $^{WT}$Coa6 structure shows a 3-helical bundle fold, where the N-terminal helical pair is tethered at each end by intramolecular disulfide bonds. We propose that in the mitochondrial IMS, the Cys58–Cys90 disulfide redox cycles between oxidized and reduced states and that when reduced, may constitute a Cu(I)-binding site. This proposal is supported by our data showing that the Coa6 protein in human mitochondria is partially reduced, that residues Cys58 and Cys90 can be labeled by IAA under reducing conditions, and that in the structure of $^{WT}$Coa6, the Cys58–Cys90 disulfide shows a geometry and surrounding protein structure that are amenable to redox cycling and copper binding, respectively. Finally, site-directed mutagenesis of residues Cys58 and Cys90 to Ser eliminates Cu(I) binding. However, the question remains as to the precise role of Coa6 in COX biogenesis. The question of whether Coa6 directly facilitates copper delivery to the COX Cu$_A$ site or mediates the activities and redox states of other proteins in the pathway remains.

What is known is that the biogenesis pathway for the COX Cu$_A$ site relies on the execution of finely tuned protein–protein interactions. The disulfide-mediated oligomerization, which occurs for the $^{W59C}$Coa6-mutant protein inhibits these interactions and, therefore, leads to the pathogenic outcome of complex IV deficiency.

During the revision of this article, Soma et al (54) reported the characterization of the solution structure of $^{WT}$Coa6 and determination of its redox properties. Consistent with our studies, they found that $^{WT}$Coa6 has a CHCH fold, determined a comparable redox potential for the protein (−330 mV, pH 7.0), and suggested that the pathogenic effects of the $^{W59C}$Coa6 variant may stem from changes in the interactions between Coa6 and other

members of the COX assembly pathway (such as Sco1 and COX2). There are some differences between the findings of the Soma et al report and the current work, however, which must await further investigation.

# Materials and Methods

Throughout this study, [WT]Coa6 refers to Coa6 isoform 3 (11), which is composed of residues 47–125 of the full-length Coa6, isoform 1 (UniProtKB: Q5JTJ3). Residue numbering for all protein constructs described here ([WT]Coa6, [W59C]Coa6, [C58S/C90S]Coa6, [C68S/C79S]Coa6, [Y97A]Coa6, [Y104A]Coa6, [R101A]Coa6, and [Y97A/Y104A]Coa6), including the structural descriptions and the submitted PDB coordinates follows the PDB convention, numbered according to the full-length Coa6, including the signal sequence. Therefore, the N-terminal residue of [WT]Coa6 is numbered Met47.

## Protein overexpression, purification, and characterization

The [WT]Coa6 and [W59C]Coa6 cDNAs encoding Coa6 isoform 1 residues 47–125 were amplified via PCR and subcloned into a pGEX-6P-1 GST fusion vector with an intervening PreScission Protease site for cleavage of the GST tag. Double cysteine mutations ([C58S/C90S]Coa6 and [C68S/C79S]Coa6) and mutations located at dimer interface ([Y97A]Coa6, [Y104A]Coa6, [R101A]Coa6, and [Y97A/Y104A]Coa6) were introduced into the ORF encoding the [WT]Coa6 protein using a Q5 Site-Directed Mutagenesis kit (New England Biolabs), according to the manufacturer's instructions. We also introduced a C58S/C90S double mutation into the ORF encoding the [WT]Coa6 protein in the pBABE-puro plasmid (Addgene) for cell culture and complementation studies using the same method.

Forward and reverse primers for all mutant proteins were designed using NEBaseChanger online tool (http://nebasechanger.neb.com/). The DNA sequence encoding all variants were individually amplified via PCR followed by ligation and template DNA removal using KLD enzyme mix (containing a kinase, a ligase, and DpnI, New England BioLabs).

The pGEX-6P-1 plasmids containing DNA sequences encoding [WT]Coa6, [W59C]Coa6, [C58S/C90S]Coa6, [C68S/C79S]Coa6, [Y97A]Coa6, [Y104A]Coa6, [R101A]Coa6, and [Y97A/Y104A]Coa6 were individually transformed into *E. coli* strain SHuffle T7 (New England Biolabs). Shuffle T7 cells are engineered *E. coli* K12, which constitutively express a chromosomal copy of the disulfide bond isomerase DsbC. DsbC promotes the correction of mis-oxidized proteins into their correct form in the cytoplasm (23, 24). Cultures were grown at 30°C in Luria Broth supplemented with ampicillin (100 $\mu$g·ml$^{-1}$), chloramphenicol (35 $\mu$g·ml$^{-1}$), and streptomycin (50 $\mu$g·ml$^{-1}$) to an $OD_{600}$ of 0.8, induced with IPTG (0.2 mM) and harvested after 16 h (with shaking) at 16°C.

GST tagged [WT]Coa6, [W59C]Coa6, [C58S/C90S]Coa6, [C68S/C79S]Coa6, [Y97A]Coa6, [Y104A]Coa6, [R101A]Coa6, and [Y97A/Y104A]Coa6 were purified by glutathione (GSH) affinity chromatography. Frozen cell pellets were thawed at room temperature and resuspended in PBS (pH 7.4). Cells were disrupted by passage through a TS series bench top cell disruptor (Constant Systems Ltd) at 35 kpsi. Cell debris was removed by centrifugation (Beckman JLA-25.50, 30,000$g$, 20 min, 4°C) and the soluble fraction was incubated with glutathione Sepharose 4B resin (GE Healthcare) equilibrated with PBS. The GST tag was cleaved with PreScission Protease followed by SEC (HiLoad 16/600 Superdex 75 pg, GE Healthcare; 20 mM Tris-MES, pH 8.0, and 150 mM NaCl). Cleavage of the N-terminal GST tag introduced five additional residues (GPLGS) to the N terminus of all proteins. The purified proteins were concentrated to 20 mg·ml$^{-1}$ before storage at −80°C.

## Ellman's assay

Purified [WT]Coa6 samples (72 $\mu$M) were added to Ellman's reaction buffer (2.5 ml; 0.1 M sodium phosphate, pH 8.0, 1 mM EDTA, and 2% wt/vol SDS) containing 50 $\mu$l of Ellman's reagent (4 mg·ml$^{-1}$) and incubated at room temperature for 15 min. A solution of Ellman's reagent produces a measurable yellow-colored product when it reacts with sulfhydryl groups. The reaction was monitored by measuring the absorbance of the resulting solutions at 412 nm. In the absence of free sulfhydryls, the Ellman's reagent exhibits no reaction and, therefore, $OD_{412}$ = 0.0 (25).

## Protein copper loading

Purified [WT]Coa6 was exchanged into buffer (20 mM Tris-MES, pH 8.0) by centrifugal ultrafiltration (Millipore) and incubated for 30 min with $CuSO_4$ and reduced GSH (molar ratio 1: 5: 10; protein: $CuSO_4$: GSH). To remove the excess Cu from the mixture, the incubated protein sample was applied to a SEC column (HiLoad 16/600 Superdex 75 pg, GE Healthcare). The presence of Cu(I) in the peak fractions was analyzed colorimetrically using the ligand bathocuproinedisulfonic acid (Bcs) and those protein fractions containing Cu were pooled and concentrated by centrifugal ultrafiltration. The Cu:protein stoichiometries of the protein samples prepared in this manner were confirmed by a colorimetric assay using Bcs (35), with protein concentration determined by a BCA assay (Pierce BCA Protein Assay Kit; Thermo Fisher Scientific) according to the manufacturer's instructions.

## Protein copper-binding experiments

Measurements of the copper-binding affinities of the [WT]Coa6 and mutant proteins ([W59C]Coa6, [C58S/C90S]Coa6, and [C68S/C79S]Coa6) were performed as previously described (11). Briefly, purified proteins (in 20 mM Tris-MES, pH 8.0) were titrated at various concentrations (1–30 $\mu$M) into solutions containing buffer (20 mM Tris-MES, pH 7.0), $CuSO_4$ (20 $\mu$M), Bcs (200 $\mu$M), and $NH_2OH$ (1 mM). The exchange of Cu(I) from the $[Cu^ICcs_2]^{3-}$ complex to the proteins was monitored by measuring the absorbance of the resulting solutions at 483 nm. The data were analyzed by plotting $[Cu^IBcs_2]^{3-}$ (as determined from the absorbance values at 483 nm) versus protein:Cu ratios, and the data fitted using the equation previously described (11, 35).

## CD spectroscopy

The secondary structures of the [WT]Coa6, [C58S/C90S]Coa6, and [C68S/C79S]Coa6 proteins were determined by CD spectroscopy in 1.0-mm path length quartz cuvettes, using an Aviv Model 420 (CD) Spectrometer. Protein samples were prepared at 0.162 mg·ml$^{-1}$ in CD

buffer (10 mM sodium fluoride, 50 mM potassium phosphate, pH 8.5, 0.5 mM tris(2-carboxyethyl)phosphine [TCEP]), and wavelength scans were recorded at 20°C between 190 and 250 nm. Data were converted to mean residue ellipticity and the secondary structure compositions of the protein samples estimated using the CDPro software with CONTINLL and CDSSTR algorithms (55).

## Analytical SEC

$^{WT}$Coa6, $^{Y97A}$Coa6, $^{Y104A}$Coa6, $^{R101A}$Coa6, and $^{Y97A/Y104A}$Coa6 proteins (100 μg) were applied to an analytical SEC column (Superdex 200 Increase 3.2/300; GE Healthcare) pre-equilibrated with 20 mM Tris-MES, pH 8.0, and 150 mM NaCl buffer and eluted at 0.05 ml·min⁻¹ in the same buffer. Elution volumes for each variant were compared with that of $^{WT}$Coa6, which was applied to and eluted from the column under the same conditions.

## MS

A sample of $^{WT}$Coa6 (200 μM) was reduced with 4 mM DTT and then alkylated with 50 mM IAA for 30 min before MALDI-TOF. In addition, IAA-labeled and unlabeled samples were subjected to trypsin digestion and tandem MS/MS sequencing using a Thermo Fisher Scientific LTQ Orbitrap Elite ETD Mass Spectrometer as previously reported (56). Upon digestion, peptides containing the free sulfhydryl groups on the cysteine residues shift by 58 Da because of the alkylation. The predicted mass of alkylated peptides were calculated using PEPTIDEMASS (57).

## Determination of redox potential of $^{WT}$Coa6

Redox potential of the S–S/SH redox couple of $^{WT}$Coa6 was determined as previously described (58). Briefly, $^{WT}$Coa6 (200 μM) was incubated at room temperature in 100 mM phosphate buffer and 1 mM EDTA (pH 7.0) containing 20 mM oxidized DTT and increasing concentrations of reduced DTT (0–800 mM). After 2-h incubation, the reactions were stopped by addition of 10% trichloroacetic acid and centrifuged (13,000$g$). The precipitated protein pellets were washed with ice-cold 100% acetone, dissolved in 4′-acetamido-4′-maleimidylstilbene-2,2′-disulfonic acid (AMS) buffer (2 mM AMS, 1% SDS, and 50 mM Tris [pH 7.0]) to label free thiols and analyzed by SDS–PAGE to separate the oxidized and AMS-bound reduced forms. The fractions of reduced protein were quantified using the ImageJ software (59) and plotted against the buffer ratio [DTT$_{Red}$]/[DTT$_{Ox}$]. The fraction of thiolate as a function of [DTT$_{Red}$]/[DTT$_{Ox}$] was plotted to calculate the equivalent intrinsic redox potential as described previously (60).

## Cell culture

HEK293T COA6$^{KO}$ cell lines were previously described (11). The cells were grown in high glucose DMEM (Invitrogen) containing 10% (vol/vol) FBS and penicillin/streptomycin and 50 μg·ml⁻¹ uridine at 37°C under an atmosphere of 5% CO$_2$. Isoform 3 of COA6 was used for complementation studies, and constructs were subcloned into pBABE-puro (Addgene) as previously described (11). Retroviral constructs along with packaging plasmids were transfected into

HEK293T cells using Lipofectamine 2000. Viral supernatant was collected at 48 h post-transfection and used to infect COA6$^{KO}$ cells in the presence of 8 μg·ml⁻¹ polybrene. Transduced cells were expanded following selection for 48 h under 1 μg·ml⁻¹ puromycin.

## BN-PAGE and Western blot analysis

Mitochondria were isolated as previously described (61). Mitochondria were solubilized in 1% digitonin and separated on 4–10% acrylamide–bisacrylamide BN-PAGE gels as previously described (62) and detected by Western blot using a total OXPHOS Rodent WB Antibody Cocktail (ab110413; Abcam).

## Immunofluorescence assay

Cells were fixed with 4% (wt/vol) paraformaldehyde in PBS (pH 7.4) for 10 min followed by permeabilization with 0.2% (wt/vol) Triton X-100 in PBS before incubation with primary antibodies against Tom20 (rabbit polyclonal, 1:500; Santa Cruz, SC11415) and Flag (mouse monoclonal, 1:100; Sigma-Aldrich, F1804-1MG) for 90 min in 3% BSA, 0.02% Tween-20 in PBS at room temperature. Primary antibodies were labeled with either Alexa Fluor488-conjugated antimouse-IgG or Alexa Fluor568-conjugated antirabbit-IgG secondary antibodies (A-11001 and A-11011; Thermo Fisher Scientific, respectively). Hoechst 33,258 (1 μg·ml⁻¹) was used to stain nuclei. Confocal microscopy was performed using a Leica TCS SP8 confocal microscope (405, 488, 552, and 647 nm; Leica Microsystems) equipped with HyD detectors. Z-sectioning was performed using 300-nm slices and combined. All images were processed using ImageJ (59).

## Protein crystallization and data collection

Crystallization trials were conducted using commercially available screens (SaltRx HT and Index HT [Hampton Research]) by sitting drop vapor diffusion in 96-well plates (Molecular Dimensions) using pure $^{WT}$Coa6 and $^{W59C}$Coa6 samples at two different protein concentrations (10 and 20 mg·ml⁻¹). Crystallization drops consisting of equal volumes (0.2 μl) of reservoir and protein solutions were dispensed using a Crystal Gryphon Liquid Handling System (Art Robbins Instruments) and were equilibrated against a reservoir of screen solution (50 μl). Plates were incubated at 20°C. Several tiny crystals of $^{WT}$Coa6 were observed within 2 wk in conditions D8 and D9 of the Index HT screen (0.1 M Hepes, pH 7.5, 25% [wt/vol] PEG 3350, and 0.1 M Tris, pH 8.5, 25% [wt/vol] PEG 3350, respectively) and multiple small $^{W59C}$Coa6 crystals were obtained after 5 d in conditions G10, G11, G6, and G7 of the Index HT screen (0.2 M magnesium chloride hexahydrate, 0.1 M Bis-Tris, pH 5.5, 25% wt/vol polyethylene glycol 3350; 0.2 M magnesium chloride hexahydrate, 0.1 M Bis-Tris, pH 6.5, 25% wt/vol polyethylene glycol 3350; 0.2 M ammonium acetate, 0.1 M Bis-Tris, pH 5.5, 25% wt/vol polyethylene glycol 3350; and 0.2 M ammonium acetate, 0.1 M Bis-Tris, pH 6.5, 25% wt/vol polyethylene glycol 3350, respectively). Optimization of these conditions was carried out by hanging-drop vapor diffusion in 24-well VDX plates (Hampton Research). Diffraction-quality crystals of $^{WT}$Coa6 grew after 7 d at 20°C by hanging-drop vapor diffusion with drops containing equal volumes (1 μl) of $^{WT}$Coa6 (30 mg·ml⁻¹ in

20 mM Tris-MES, pH 8.0, and 150 mM NaCl) and crystallization solution (0.1 M Hepes, pH 7.6, and 29% [wt/vol] PEG 3350) equilibrated against 500 $\mu$l reservoir solution. $^{W59C}$Coa6 was crystallized at 20°C by hanging-drop vapor diffusion with drops consisting of equal volumes (1 $\mu$l) of protein (25 mg·ml$^{-1}$ in 20 mM Tris-MES, pH 8.0, and 150 mM NaCl) and reservoir solution (0.1 M Bis-Tris, pH 5.7, 0.23 M MgCl$_2$·6H$_2$O, and 29% [wt/vol] PEG 3350). Both $^{WT}$Coa6 and $^{W59C}$Coa6 crystals were cryoprotected in reservoir solution containing 25% (vol/vol) glycerol before flash-cooling in liquid nitrogen.

Diffraction data were recorded on an ADSC Quantum 315r detector at the Australian Synchrotron on beamline MX2 at wavelengths of 0.9918, 1.722 and 0.954 Å for the $^{WT}$Coa6 native data set, $^{WT}$Coa6 anomalous data set and $^{W59C}$Coa6, respectively. All data were collected at 100 K and were processed and scaled with HKL2000 (63). Data collection statistics are detailed in Table 1.

### Structure solution and refinement

The crystal structure of $^{WT}$Coa6 was determined by sulfur single-wavelength anomalous dispersion. Eight sulfur sites were identified using SHELXD and phasing performed by SHELXE within the CCP4 suite (64). Statistical phase improvement and solvent flattening was carried out using the program PIRATE from the CCP4 suite (64). Initial model building was carried out using BUCCANEER from the CCP4 suite (64) with manual model building and the addition of water molecules were carried out in COOT (65). The model was refined using REFMAC5 (66). The quality of the structure was determined by MOLPROBITY (67) (Table 1).

The crystal structure of $^{W59C}$Coa6 was solved by molecular replacement using the program PHASER (68) from the CCP4 suite (64). The crystal structure of $^{WT}$Coa6 was used as a search model after removal of all water molecules. The model was refined using REFMAC5 (66) and manual model building and the addition of water molecules were carried out in COOT (65). The quality of the structure was determined by MOLPROBITY (67) (Table 1).

Structure superpositions and the calculation of r.m.s.d's were carried out with LSQKAB (as part of the CCP4 suite) (65) and analyses of oligomeric protein complexes and buried surface areas were performed with the PDBePISA server (69).

# Supplementary Information

# Acknowledgements

This study was funded by the Australian Research Council (DP140102746 to MJ Maher), the National Health and Medical Research Council (GNT1165217 to MT Ryan and MJ Maher; GNT1140851 to DA Stroud), and an Australian Government Research Training Program Scholarship to S Maghool. Aspects of this research were undertaken on the Macromolecular Crystallography beamline MX2 at the Australian Synchrotron (Victoria, Australia) and we thank the beamline staff for their enthusiastic and professional support. We also acknowledge the La Trobe University Comprehensive Proteomics Platform for providing infrastructure for this study.

## Author Contributions

S Maghool: data curation, formal analysis, investigation, and writing—original draft, review, and editing.
NDG Cooray: investigation.
DA Stroud: conceptualization, data curation, formal analysis, supervision, funding acquisition, investigation, and writing—review and editing.
D Aragão: data curation, investigation, methodology, and writing—review and editing.
MT Ryan: conceptualization, formal analysis, supervision, funding acquisition, and writing—review and editing.
MJ Maher: conceptualization, data curation, formal analysis, supervision, funding acquisition, validation, investigation, project administration, and writing—original draft, review, and editing.

## Conflict of Interest Statement

The authors declare that they have no conflict of interest.

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
