## [Reviewer comments · Life Science Alliance]

Life Science Alliance

Structural and functional characterization of the mitochondrial complex IV assembly factor Coa6

Shadi Maghool, N. Dinesha Cooray, David Stroud, David Aragão, Michael Ryan, and Megan Maher
DOI: <https://doi.org/10.26508/lsa.201900458>

Corresponding author(s): Megan Maher, The University of Melbourne

Review Timeline:	Submission Date:	2019-06-17
	Editorial Decision:	2019-07-09
	Revision Received:	2019-08-05
	Editorial Decision:	2019-08-06
	Revision Received:	2019-08-26
	Accepted:	2019-09-02

Scientific Editor: Andrea Leibfried

Transaction Report:

July 9, 2019

Re: Life Science Alliance manuscript #LSA-2019-00458

Dr. Megan Maher
La Trobe University
La Trobe Institute for Molecular Science
Kingsbury Drv
Melbourne, Victoria 3086
Australia

Dear Dr. Maher,

Thank you for submitting your manuscript entitled "Structural and functional characterization of the mitochondrial complex IV assembly factor Coa6" to Life Science Alliance. The manuscript was assessed by expert reviewers, whose comments are appended to this letter.

As you will see, the reviewers appreciate the structural insight provided into Coa6 and they provide constructive input on how to further strengthen your work. We would thus like to invite you to submit a revised version of your manuscript to us, addressing the concerns raised by rev#2 and #3. Importantly, more definitive insight into copper binding should get provided (rev#2) and the oxidation state of other cysteines should get checked (other points 1 and 2 of this reviewer). The two last points raised by rev#2 and the comments of rev#3 can get addressed by discussion/restructuring.

Thank you for this interesting contribution to Life Science Alliance. We are looking forward to receiving your revised manuscript.

Sincerely,

B. MANUSCRIPT ORGANIZATION AND FORMATTING:

Reviewer #1 (Comments to the Authors (Required)):

The manuscript "Structural and functional characterization of the mitochondrial complex IV

assembly factor Coa6" describes the first structure of Coa6 protein involved in CIV maturation. Structures of WT and mutant proteins are of high quality and suggest a possible site of Cu binding. Although how exactly Coa6 participates in CIV assembly is still not clear, this work is a significant step towards this goal. The work is technically solid and the manuscript is clear and well written.

Reviewer #2 (Comments to the Authors (Required)):

This manuscript presents the crystal structures of the human wild-type Coa6 and the pathogenic W59C Coa6 mutant. The copper binding properties of wild-type Coa6 were also investigated. The wild-type Coa6 structure shows a three helical bundle fold, where the N-terminal helical pair is tethered by two intramolecular disulfide bonds. On the basis of indirect experimental data, the authors propose that Cys58 and Cys90, which form a disulfide bond in the crystal structure, bind a copper(I) ion upon its specific disulfide bond reduction.

The crystal structure of the pathogenic W59C Coa6 mutant shows that the protein form a dimer of dimers mediated by a disulfide bond involving Cys 59 from two protein molecules. This disulfide mediated oligomerization of the W59C Coa6 protein is proposed to provide a structural explanation for this loss of function mutation.

This work is well done and the data clearly presented. My only main concern is related to the data describing the copper binding site. The experiments supporting that Cys58 and Cys90 are the best candidate to bind the copper(I) ion are convincing, but not definitive. A direct experiment is required to assess this proposal. I strongly suggest to apply ¹⁵N-edited NMR to directly identify the copper(I) binding site.

Other points:

1. When the authors produced purified apo wild-type Coa6 (at pg.6, line 163-164), are all the four cysteines reduced or only the Cys58-Cys90 disulfide is reduced? This needs to be checked.
2. Peptide analysis by tandem mass spectrometry shows the reduction of the Cys58-Cys90 disulfide bond. What about Cys68 and Cys79? Do they result oxidized by tandem mass spectrometry?
3. Examination of the cellular localization of the overexpressed C58S/C90S Coa6-FLAG protein by immunofluorescence showed that unlike WT Coa6-FLAG, the C58S/C90SCoa6 protein was cytosolic. This strongly suggests that the Mia40-mediated oxidation of these cysteines to form an intramolecular disulfide bond is required to trap the protein into the IMS, as already demonstrated for other CHCH IMS proteins (see Proc Natl Acad Sci U S A. 2010 Nov 23;107(47):20190-5, this work should be cited). This result presented by the authors presupposes that this disulfide needs to be re-reduced in vivo in order to bind the copper(I) ion once the protein is imported in the IMS. But which is the system in the IMS able to selectively reduce this disulfide bond? Is the reduction potential of this disulfide higher than the reduction potential of the CX9C intramolecular disulfides of Cox17 reported by the authors at pg. 7, lines 179-180 and higher of the other disulfide of Coa6?
4. The authors assess that the observed changes in Cu(I) binding properties of the mutants with respect to the wild-type protein were not due to protein misfolding on the basis of CD spectra. However, CD monitors only secondary structural changes, and the removal of a disulfide bond can largely modify the tertiary structural arrangement of the two helices without significantly affecting the secondary structure. This has been shown to occur, indeed, in the CHCH protein Cox17 (see J Biol Chem. 2011 Sep 30;286(39):34382-90). These aspects need to be discussed.

Reviewer #3 (Comments to the Authors (Required)):

The manuscript by Maghool and coworkers describes the structure of the cytochrome c oxidase assembly protein Coa6. The manuscript describes the dimeric structure highlights the similarities in the fold to other proteins of the same class and shows nicely the possible copper binding residues. The manuscript could be improved by including a docking simulation between the Coa6 structure and any of the partner proteins that have structures available i.e. COX2, SCO1, SCO2. These simulations suggesting important interface residues for further functional studies would enhance the advance offered to the field. In addition a comparison to surface of the disease causing mutation would be informative. While these description may be possible from access to the pdb they will enhance the discussion of the structure presented here.

The structural data is well supported and this is the first structure of this assembly factor. The table (or other figure) supporting the MS data for reduced cysteine residues in the protein could be added to the main text as this is a very important point to support the copper binding studies, the status of the cysteine residues in these classes of proteins in vivo remains one of the most challenged and challenging aspects to understanding the function of these assembly factors. The copper binding/competition studies are well defined and appropriate. It should be possible to create a chimeric protein with an inner membrane tether as was done previously with COX19 to generate mitochondrially localized version of the mutants to test for functionality this would require extensive studies to verify localization and function so this could be considered for future studies.

Overall the conclusions of the manuscript are supported by the data presented and advance our knowledge of Coa6. But the manuscript would be enhanced by discussion of the interaction interfaces particularly if place in context of the variation in in vivo interactions observed in multiple laboratories.

Reviewer 1.

The manuscript "Structural and functional characterization of the mitochondrial complex IV assembly factor Coa6" describes the first structure of Coa6 protein involved in CIV maturation. Structures of WT and mutant proteins are of high quality and suggest a possible site of Cu binding. Although how exactly Coa6 participates in CIV assembly is still not clear, this work is a significant step towards this goal. The work is technically solid and the manuscript is clear and well written.

We thank Reviewer 1 for these comments – there is nothing to address.

Reviewer 2.

1. *This work is well done and the data clearly presented. My only main concern is related to the data describing the copper binding site. The experiments supporting that Cys58 and Cys90 are the best candidate to bind the copper(I) ion are convincing, but not definitive. A direct experiment is required to assess this proposal. I strongly suggest to apply ¹⁵N-edited NMR to directly identify the copper(I) binding site.*

Our preliminary undertakings using NMR approaches, have revealed issues with the discrimination of allosteric effects *versus* copper binding to Coa6. There also exists the possibility that resonances from copper binding site residues and the neighbouring environment are overlapped. In essence we feel that we have used the correct approaches to identify the copper binding site in Coa6. However, we recognize the absence of structural information on Cu(I)-^{WT}Coa6. We have therefore revised the manuscript to properly reflect this uncertainty and have explicitly detailed the need for definitive structural information in future studies:

"Taken together, the redox potential of the ^{WT}Coa6 Cys58-Cys90 disulfide, the results of mutagenesis, Cu(I) binding experiments and analyses of the geometries of the intramolecular disulfide bonds of ^{WT}Coa6 indicate that the ^{WT}Coa6 Cys58-Cys90 disulfide may redox cycle and in the reduced state (2SH), bind Cu(I). The redox potential of the ^{WT}Coa6 Cys58-Cys90 S-S/2SH redox couple at -349 ± 1 mV indicates ^{WT}Coa6 could reduce the disulfide bonds in COX2 (-288 ± 3 mV) [22], and Sco1 (-277 ± 3 mV) [38] and could either reduce or be reduced by Sco2 (<-300 mV) [9]. In fact, a recent examination of this pathway suggested ^{WT}Coa6 may play such a role [22]. Unfortunately, despite extensive attempts, we were unable to crystallize the Cu(I)-bound ^{WT}Coa6. Confirmation of Cu(I) binding and the atomic details of the Cu(I) site structure therefore await future investigation."

2. *When the authors produced purified apo wild-type Coa6 (at pg.6, line 163-164), are all the four cysteines reduced or only the Cys58-Cys90 disulfide is reduced? This needs to be checked.*

The Coa6 proteins were overexpressed in *Escherichia coli* strain SHuffle[®] T7 (New England, BioLabs). Shuffle T7 cells are engineered *E. coli* K12 which constitutively express a chromosomal copy of the disulfide bond isomerase DsbC. DsbC promotes the correction of mis-folded proteins, including incorrectly paired disulfide bonds into their correct form in the bacterial cytoplasm. The oxidation state of ^{WT}Coa6 was confirmed on purification using an Ellman reaction, which gave no absorbance at 412 nm, indicating the absence of free sulfhydryl groups. This was confirmed by the crystal structure of ^{WT}Coa6, which shows two disulfide bonds – between Cys58-Cys90 and Cys68-Cys79. Our suggestion is that Cu(I) binding to ^{WT}Coa6 occurs upon the reduction of the Cys58-Cys90 disulfide bond. We have added additional text to the Materials and Methods and the Results and Discussion to clarify this point:

“Recombinant ^{WT}Coa6 protein was overexpressed in Escherichia coli strain SHuffle[®] T7 which promotes the production of correctly disulfide bonded active proteins within the cytoplasm [23, 24]. The fully oxidized ^{WT}Coa6 protein (including two disulfide bonds) was purified by affinity and size-exclusion chromatography (SEC). The redox state of the purified ^{WT}Coa6 protein was confirmed by Ellman assay [25].”

“The sequence of ^{WT}Coa6 includes four Cys residues (at positions 58, 68, 79 and 90) in a CX₉C–CX₁₀C sequence motif, which in the crystal structure form two intramolecular disulfide bonds per monomer, between Cys58–Cys90 and Cys68–Cys79, in accordance with the fully oxidized state of the protein (Fig 1A). These disulfides tether helices α_1 and α_2 together at each end of the helical pair.”

3. *Peptide analysis by tandem mass spectrometry shows the reduction of the Cys58–Cys90 disulfide bond. What about Cys68 and Cys79? Do they result oxidized by tandem mass spectrometry?*

We were not able to observe the peptides that include the residues Cys68 and Cys79 in the oxidised or reduced (and iodoacetamide labelled) protein. We have therefore added an additional mass spectrometry analysis of the intact protein. Under reducing conditions and in the presence of iodoacetamide, the mass of the dominant protein species in solution indicates labelling of two Cys residues (not four). This, together with the peptide analysis indicates the reduction of the Cys58–Cys90 disulfide bond only under the conditions of the experiment. This is detailed in the text and in Tables 2, S1 and Figure S5.

“...the purified protein was incubated with DTT_{Red}/DTT_{Ox} (40:1), followed by labeling with iodoacetamide prior to analyses by both MALDI-TOF mass spectrometry (Fig S5, Table S1) and tryptic digest and peptide analysis by tandem mass spectrometry (MS/MS; Table 2). The MALDI-TOF data revealed that the dominant species in the reduced ^{WT}Coa6 sample included two free thiol groups (2SH (not 4SH), Fig S5, Table S1). The MS/MS data showed that peptides that included residues Cys58 and Cys90 yielded molecular weights corresponding to adducts with iodoacetamide, indicating reduction of the Cys58–Cys90 disulfide bond under these conditions (Table 2).”

4. *Examination of the cellular localization of the overexpressed C58S/C90S Coa6-FLAG protein by immunofluorescence showed that unlike WT Coa6-FLAG, the C58S/C90SCoa6 protein was cytosolic. This strongly suggests that the Mia40-mediated oxidation of these cysteines to form a intramolecular disulfide bond is required to trap the protein into the IMS, as already demonstrated for other CHCH IMS proteins (see Proc Natl Acad Sci U S A. 2010 Nov 23;107(47):20190-5, this work should be cited). This result presented by the authors presupposes that this disulfide needs to be re-reduced in vivo in order to bind the copper(I) ion once the protein is imported in the IMS. But which is the system in the IMS able to selectively reduce this disulfide bond? Is the reduction potential of this disulfide higher than the reduction potential of the CX₉C intramolecular disulfides of Cox17 reported by the authors at pg. 7, lines 179-180 and higher of the other disulfide of Coa6?*

We have now cited the reference as recommended and revised this section of the text accordingly. We have also measured the redox potential of the Cys58–Cys90 disulfide as -349 ± 1 mV (pH 7.0), which is comparable to that reported for the Sco2 protein. Redox cycling of Coa6 therefore may be mediated by Sco2 or another as yet unidentified protein in this pathway. A discussion of these analyses has been added to the manuscript (see text below and that detailed under point 1, above).

“CX₉C–CX₁₀C and CX₉C proteins require the presence of the Cys residues within these motifs for import into the IMS, and oxidative folding via Mia40 to trap the proteins in the IMS [49, 50]. The

absence of IMS localization of the ^{C58S/C90S}Coa6-FLAG protein correlates with these observations and impeded our efforts to probe the role of these residues in Coa6 function and specifically COX assembly in human cells.”

5. The authors assess that the observed changes in Cu(I) binding properties of the mutants with respect to the wild-type protein were not due to protein misfolding on the basis of CD spectra. However, CD monitors only secondary structural changes, and the removal of a disulfide bond can largely modify the tertiary structural arrangement of the two helices without significantly affecting the secondary structure. This has been shown to occur, indeed, in the CHCH protein Cox17 (see J Biol Chem. 2011 Sep 30;286(39):34382-90). These aspects need to be discussed.

We have now included a discussion of these aspects in the revised manuscript as suggested.

“Circular dichroism spectroscopic analyses of the mutant proteins showed that both proteins gave spectra similar to that of the ^{WT}Coa6 protein, indicating that the observed changes in Cu(I) binding properties were not due to alterations in the secondary structures of the proteins in the presence of the mutations (Fig S6). However, the possibility exists that the mutagenesis affected the flexibility of the protein and/or its tertiary structure as observed for a similar analysis of the human Cox17 protein [42].”

Reviewer 3.

1. The manuscript could be improved by including a docking simulation between the Coa6 structure and any of the partner proteins that have structures available i.e. COX2, SCO1, SCO2. These simulations suggesting important interface residues for further functional studies would enhance the advance offered to the field. In addition a comparison to surface of the disease causing mutation would be informative. While these description may be possible from access to the pdb they will enhance the discussion of the structure presented here.

We have analysed the electrostatic surfaces of the relevant structures and included this discussion in the manuscript. We have introduced an additional figure (Fig S9) to illustrate these elements.

“The structure of ^{W59C}Coa6 is consistent with the bulk of these data. Oligomerization of the protein, through the creation of intermolecular disulfide bonds may inhibit or eliminate targeted protein-protein interactions between Coa6 and proteins such as Sco1, Sco2 and COX2, which are critical for its function. Certainly, the structures and charge distributions on the surfaces of the ^{WT}Coa6 dimer and ^{W59C}Coa6 tetramer are different (Fig S9A,B). In particular, condensed areas of positive charge on the surface of the ^{WT}Coa6 dimer (Fig S9A) are not present in the ^{W59C}Coa6 tetramer (Fig S9B). These regions may mediate interactions with the Sco1, Sco2 and COX2 proteins through concentrated areas of negative charge on the surface structures of those proteins (Fig S9C,D,E).”

2. The table (or other figure) supporting the MS data for reduced cysteine residues in the protein could be added to the main text as this is a very important point to support the copper binding studies, the status of the cysteine residues in these classes of proteins in vivo remains one of the most challenged and challenging aspects to understanding the function of these assembly factors.

This table has been moved to the main text in the revised manuscript (Table 2).

August 6, 2019

RE: Life Science Alliance Manuscript #LSA-2019-00458R

Dr. Megan Maher
The University of Melbourne
School of Chemistry and The Bio21 Molecular Science and Biotechnology Institute
Flemington Rd
Parkville, Victoria 3010
Australia

Dear Dr. Maher,

Thank you for submitting your revised manuscript entitled "Structural and functional characterization of the mitochondrial complex IV assembly factor Coa6". I have now assessed the changes introduced in revision and think that they address the reviewer concerns well. I also appreciate that you attempted to further test for copper binding and think that this reviewer concern is sufficiently addressed at this stage given the difficulties you encountered. We would thus be happy to publish your paper in Life Science Alliance pending final revisions necessary to meet our formatting guidelines:

- Fig3: the data fitting seems missing for C58S/C90S variant, please fix
- Please add callouts in the manuscript text for Fig2B and 4C
- Please link your profile in our submission system to your ORCID iD, you should have received an email with instructions on how to do so

A. FINAL FILES:

B. MANUSCRIPT ORGANIZATION AND FORMATTING:

Sincerely,

September 2, 2019

RE: Life Science Alliance Manuscript #LSA-2019-00458RR

Dr. Megan Maher
The University of Melbourne
School of Chemistry and Bio21 Institute
30 Flemington Rd
Parkville, Victoria 3010
Australia

Dear Dr Maher,

Thank you for submitting your Research Article entitled "Structural and functional characterization of the mitochondrial complex IV assembly factor Coa6". It is a pleasure to let you know that your manuscript is now accepted for publication in Life Science Alliance. Congratulations on this interesting work.

*****IMPORTANT:** If you will be unreachable at any time, please provide us with the email address of an alternate author. Failure to respond to routine queries may lead to unavoidable delays in publication.*******

DISTRIBUTION OF MATERIALS:

Again, congratulations on a very nice paper. I hope you found the review process to be constructive and are pleased with how the manuscript was handled editorially. We look forward to future exciting submissions from your lab.

Sincerely,

Daniel Klimmeck

Daniel Klimmeck, PhD
Scientific Editor
Life Science Alliance